# Coutilisation of oral rehydration solution and zinc for treating diarrhoea and its associated factors among under-five children in East Africa: a multilevel robust Poisson regression

Bruck Tesfaye Legesse [ID],[1] Wubet Tazeb Wondie [ID],[2]
Gezahagn Demsu Gedefaw [ID],[3] Yakob Tadese Workineh,[4]
Beminate Lemma Seifu [ID][5]

For numbered affiliations see end of article.

**Correspondence to**
Bruck Tesfaye Legesse;
brucktesfaye143@gmail.com

## ABSTRACT

**Objective** This study aimed to assess the coutilisation of oral rehydration solution (ORS) and zinc for treating diarrhoea and its associated factors among under-5 children in East Africa.

**Design** Cross-sectional study design. Multilevel Poisson regression analysis with robust variance was fitted to identify predictors of zinc and ORS coutilisation. An adjusted prevalence ratio (aPR) with a 95% CI was reported to declare the statistical significance.

**Setting** Twelve East African countries.

**Participants** 16 850 under-5 children who had diarrhoea were included in the study.

**Result** In East African nations, the coutilisation of ORS and zinc for the treatment of diarrhoea in children under 5 was 53.27% with a 95% CI (52.54% to 54.01%). Children of mothers with primary education (aPR 1.15, 95% CI 1.09 to 1.20), secondary education (aPR 1.08, 95% CI 1.02 to 1.14), higer education (aPR 1.19, 95% CI 1.10 to 1.29), those from maternal age category of 20–24 (aPR 1.14, 95% CI 1.07 to 1.21), age category of 25–29 (aPR 1.13, 95% CI 1.06 to 1.21), age category of 30–34 (aPR 1.09, 95% CI 1.02 to 1.16), those from wealthy households (aPR 1.04, 95% CI 1.01 to 1.09) and those who have a media exposure (aPR 1.04, 95% CI 1.01 to 1.08) were more likely to receive combination.

**Conclusion** Only half of the under-5 children with diarrhoea in East Africa were treated with a combination of ORS and zinc. To increase the use of the suggested combination therapy of ORS with zinc, it is important to empower women through education and prevent teen pregnancy.

## STRENGTHS AND LIMITATIONS OF THIS STUDY

⇒ This study was based on nationally representative large data and used appropriate statistical analysis techniques.

⇒ Despite the aforementioned strengths, the measure of zinc utilisation practice was based on the mother's recall, which could lead to recall bias.

⇒ Second, the Demographic and Health Survey (DHS) lacks information on zinc availability and the role of diet in zinc intake. As a result, we are unable to evaluate these factors.

⇒ Some East African countries are not included in the DHS programme. Furthermore, due to the cross-sectional nature of the data, a clear temporality (cause-and-effect relationship) between the dependent and independent variables was not established.

## BACKGROUND

Diarrhoea is three or more loose or watery stools in a 24-hour period, which can also present as a change in the volume, frequency or fluidity of faeces.[1] Although there are easy ways to prevent and cure diarrhoea in children, it continues to be the world's greatest cause of mortality and undernutrition in children under the age of 5.[2] Each year, around 525 000 children under the age of 5 lose their lives to diarrhoea,[3] and the data taken from the public WHO dataset, which included information from 195 nations between 2000 and 2017 findings showed that the largest frequency of diarrhoea-related mortality was found in Asian and African nations. The results of the spatial modelling also showed that, up until 2010, the majority of diarrhoea-related deaths occurred in Asian nations; however, in 2011, the majority of deaths moved to Africa.[2] In East Africa, diarrhoeal-related under-5 mortality accounts from 2.1% in Tanzania,[4] 8% in Mozambique,[5] 8.8% in Ethiopia,[6] 9% in Kenya[7] and 18.2% in Uganda.[4]

One of the top priorities for reducing unnecessary child deaths under the Sustainable Development Goal is treating diarrhoea.[8] Up to 93% of diarrhoea-related death can be avoided using oral rehydration

solution (ORS) and zinc, two very affordable and extremely effective diarrhoea therapies. Because of this, the WHO and the UNICEF suggest combining ORS with zinc for a successful therapy.[9 10] Copackaging zinc with ORS has the potential to promote their combined usage, improve treatment access and maximise treatment use.[11] The combination of zinc and ORS in a visually pleasing copack with counselling messages and instructions greatly enhanced healthcare providers' prescribing practices and parents' compliance with zinc administration.[12] Another study conducted in Cambodia reached the same conclusion: copackaging is a useful strategy for promoting the combined utilisation of ORS and zinc.[13]

Zinc supplementation lessens the length and intensity of episodes and lowers the likelihood of recurrence in the short term,[14 15] ORS contains trisodium citrate dehydrate, glucose, sodium chloride and potassium chloride which is important to replenishes the vital fluids and salts lost via diarrhoea.[16] A study conducted on zinc supplementation at a Dhaka hospital (International Centre for Diarrhoeal Disease Research, Bangladesh) indicated that dehydration, bloody diarrhoea and fever were less with those under 5 children received a zinc for diarrhoea at home.[17]

However, the global prevalence of using ORS with zinc to treat diarrhoea continues to be alarmingly low.[2] Less than 7% of children under the age of 5 in low-income and middle-income countries (LMICs) receive zinc treatment, according to a 2017 UNICEF report, and only 44% of them receive ORS.[18] Access to this life-saving medicine might be made easier in many of these nations, many of which have high mortality rates from diarrhoea.[19]

In East Africa, only a small percentage of under-5 children receive effective treatment or ORS with Zinc for diarrhoeal disease. A study in Kenya mentioned that children treated with bundled ORS with zinc accounts for 15%,[20] while another study in Ethiopia found only 16.65%.[21] Another study conducted in Uganda mentioned that 30% of those under 5 children treated for diarrhoea took copacked ORS with zinc.[22]

According to studies done so far, maternal education, economic status, media exposure, residency, proximity to a medical facility and health insurance coverage have all been found to influence the combination of Zinc and ORS for the treatment of diarrhoea in under 5 children.[21 23–25]

Despite the clinical significance of using ORS and zinc together, and the recommendation of WHO and UNICEF for coutilisation of ORS and zinc, there is limited evidence of no research that ascertains the prevalence and contributing variables of ORS and zinc co-use among young children with diarrhoea in East Africa. So, in order to bridge this gap, a study was conducted. Since this study is the first of its kind in East Africa, policymakers and those making decisions may potentially use it as a starting point when deciding how best to encourage uptake.

## METHODS
### Data source, study setting and population
This study was based on the most recent Demographic and Health Survey (DHS) data from 2011 to 2022 of 12 East African countries. Those countries are Burundi, Ethiopia, Kenya, Comoros, Madagascar, Malawi, Tanzania, Uganda, Zambia and Zimbabwe. DHS is a nationally representative survey routinely conducted every 5 years and collects data on basic health indicators like mortality, morbidity, fertility, and maternal and child health-related characteristics. The study participants were selected using a two-stage stratified sampling technique used for the survey. In the first stage, enumeration areas (EAs) were randomly selected based on the country's recent population, and using the housing census as a sampling frame, households were randomly selected in the second stage. Different datasets, including those for men, women, children, births and households, are included in each nation's survey. For this study, the study population was under-5 children, thus, we used the kid's record dataset (KR file). All children under-5 children in East Africa who had diarrhoea were the study's source population. All under-5 children who had diarrhoea in the 2 weeks before the survey were the study population. In the current study, a weighted sample of 16 850 under-5 children was considered for final analysis. Details about DHS methodology can be accessed at https://dhsprogram.com/Methodology/index.cfm.

### Study variables and definitions
#### Dependent variable
The coutilisation of ORS and zinc was dichotomised as 'yes=1' if the child uses both ORS and zinc for the treatment of childhood diarrhoea and 'no=0' if the child does not use both ORS and zinc for the treatment of childhood diarrhoea.

#### Independent variables
The independent variables were classified as community and individual-level variables. Individual-level factors were maternal age, educational status of the mother, father's educational status, working status of the mother, wealth status, media exposure, sex of household head, health insurance and sex of the child. Distance to health facility, place of residence (urban or rural), community women's education and community poverty level were considered as community-level factors.

### Operational definition
Media exposure was created from three variables (frequency of listening to the radio, watching television and reading newspapers or magazines). In this study, women who listened to radio watched television, or read newspapers/magazines at least less than once a week were considered as having exposure to media (coded 'yes') and otherwise labelled as not having media exposure (coded 'no').

Community women's education defined as the proportion of women in the cluster who pursued education at the primary, secondary and higher education levels. The aggregate of individual women's primary, secondary and higher educational attainment can show the overall educational status of women within the cluster. They were categorised into two categories a higher proportion of women's education within the cluster and a lower proportion of women's education based on the national median value.

Community poverty status defined as the proportion of poor and poorest mothers within the cluster also can be defined as the household wealth index for each child. For each cluster, proportion of poor and poorest was aggregated and showed overall poverty status within the cluster. It was categorised into two categories based on the national median value a higher proportion of poor/poorest mothers and a lower proportion of mothers within a cluster.

## Data management and statistical analysis
Data extraction, coding and analysis were done using Stata V.17 statistical software. To restore the data's representativeness, an analysis was conducted using the weighted data. Considering the hierarchical structure of the DHS data, the clustering effect was assessed by estimating the intraclass correlation coefficient (ICC). ICC helps to determine whether there is a clustering effect present in the DHS data, which has a hierarchical structure. It is important to account for any clustering effect in the model to ensure accurate analysis. A significant clustering effect was observed according to the ICC (ICC>10%). Multilevel models are a statistical technique that is used to analyse data with hierarchical or clustered structures. Traditional multiple regression techniques treat observations as independent, which can lead to inaccurate results. In contrast, multilevel models account for the hierarchy of the data, which results in more accurate standard errors for regression coefficients. This makes them a powerful tool for analysing data with complex structures. In statistical analysis, the act of ignoring clustering effects can lead to an overstatement of statistical significance. Our study has demonstrated that when studying the coutilisation of individuals, multilevel models can provide a reliable approach to identifying cluster-level effects based on country or EA. By leveraging such models, we can better understand the nature of clustering and its impact on statistical significance, thereby ensuring the validity of our results. When the outcome is prevalent in the population, such as in the case where more than 10% of the population has coutilisation of ORS and zinc, the prevalence ratio (PR) is considered the preferred method of measuring the association between exposure and outcome. The PR provides a direct measure of how much more or less likely individuals with the exposure are to have the condition compared with those without the exposure. In our study, the OR should not be used, as it is only applicable for rare events. Using OR in our

study may lead to overestimation of the effect sizes. This study was a cross-sectional study and the prevalence of coutilisation of ORS and zinc was greater than 10%, and if we report the OR, it could overestimate the association between coutilisation of ORS and zinc and the independent variables. As the PR is the best measure of association in these situations, multilevel Poisson regression analysis with robust variance was fitted to identify predictors of zinc and ORS coutilisation among under-5 children with diarrhoea. In the bivariable multilevel Poisson regression analysis, variables having a $p < 0.2$ were considered for the multivariable analysis.

For the multilevel Poisson regression analysis, four models were constructed. The first model aimed to ascertain the degree of cluster variation in the coutilisation of zinc and ORS and it was a null model devoid of explanatory factors. The second model was fitted with individual-level variables, the third with community-level variables and the fourth with both individual and community-level variables at the same time. Because the models were nested, the deviance −2log-likelihood ratio was used to compare them, and the model with the lowest deviance was the best-fitted model for the data. Finally, the 95% CI for the adjusted PR (APR) was determined, and factors in the multivariable analysis with a $p < 0.05$ were deemed significant predictors of the coutilisation of zinc and ORS among under-5 children with diarrhoea.

## Patient and public involvement
In this study, the study participants and/or the public were not directly involved in the design, conduct, reporting and dissemination of this work.

# RESULTS
## Sociodemographic characteristics
A total weighted sample of 16 850 under-5 children who had diarrhoea within 2 weeks preceding the survey were included in this study. More than half, 8965 (53.20%) of the children were males while the vast majority (79.10%) of them resided in rural areas. The majority, 7778 (46.16%) were from poor households and 3666 (21.76%) had mothers who did not have a formal education. Only 818 (5.72%) were covered by health insurance (table 1).

## The prevalence of ORS and zinc coutilisation for diarrhoea treatment
The pooled prevalence of coutilisation of ORS and zinc for the treatment of diarrhoea among under-5 children in East African countries was 53.27% with a 95% CI (52.54% to 54.01%). In East African countries, the prevalence of ORS and zinc coutilisation ranges from 24.36% (95% CI 21.85% to 26.86%) in Madagascar to 69.32% (95% CI 67.76% to 70.87%) in Malawi (figure 1).

## Random effect and model comparison
To evaluate the random or clustering effect using the ICC, we used deviance for model comparison. The null model's

**Table 1** Sociodemographic and economic characteristics of study participants in East Africa from 2011 to 2022

| Variable | Weighted frequency | Per cent |
|---|---|---|
| Individual level variables | | |
| Sex of the child | | |
| Male child | 8965 | 53.2 |
| Female child | 7885 | 46.8 |
| Age of the mother | | |
| 15–19 | 1433 | 8.51 |
| 20–24 | 4715 | 27.98 |
| 25–29 | 4545 | 26.97 |
| 30–34 | 3228 | 19.16 |
| 35–39 | 1965 | 11.66 |
| 40–44 | 757 | 4.49 |
| 45–49 | 206 | 1.22 |
| Marital status | | |
| Never in union | 965 | 5.73 |
| Married/living together | 14160 | 84.03 |
| Divorced/widowed/ separated | 1725 | 10.24 |
| Maternal education | | |
| No formal education | 3666 | 21.76 |
| Primary | 8544 | 50.71 |
| Secondary | 3942 | 23.4 |
| Higher | 697 | 4.14 |
| Father's education (n=14164) | | |
| No formal education | 2667 | 18.83 |
| Primary | 6842 | 48.31 |
| Secondary | 3657 | 25.82 |
| Higher | 997 | 7.04 |
| Maternal current working status | | |
| Not employed | 5747 | 34.11 |
| Employed | 11103 | 65.89 |
| Sex of household head | | |
| Male household head | 12778 | 75.84 |
| Female household head | 4072 | 24.16 |
| Household wealth status | | |
| Poor | 7778 | 46.16 |
| Middle | 3218 | 19.1 |
| Rich | 5854 | 34.74 |
| Covered by health insurance (n=14291) | | |
| No | 13472 | 94.28 |
| Yes | 818 | 5.72 |
| Media exposure | | |

Continued

**Table 1** Continued

| Variable | Weighted frequency | Per cent |
|---|---|---|
| No | 5959 | 35.36 |
| Yes | 10891 | 64.64 |
| Community level variables | | |
| Distance to health facility (n=15498) | | |
| Big problem | 6926 | 44.69 |
| Not a big problem | 8572 | 55.31 |
| Residence | | |
| Urban | 3521 | 20.9 |
| Rural | 13329 | 79.1 |
| Community educational status | | |
| Low | 2996 | 17.78 |
| High | 13854 | 82.22 |
| Community poverty level | | |
| Low | 7695 | 45.67 |
| High | 9155 | 54.33 |

ICC of 0.145 indicated that unmeasured or unmeasurable factors (random effects) accounted for about 14.5% of the variance in the coutilisation of zinc and ORS. In terms of comparing models, the best-fitted model was the final model, which had a lower deviance (table 2).

### Factors associated with coutilisation of ORS and zinc for under-5 diarrhoea treatment

A multilevel bivariable analysis was done to assess the crude relationship between the independent variables and ORS-zinc coutilisation. Among the variables included in the bi-variable analysis only maternal age, maternal education status, wealth index, maternal employment status, media exposure, community-level illiteracy and community-level poverty showed a statistically significant association with ORS and zinc coutilisation at a p<0.20.

In the final model, maternal age, maternal education status, wealth index and media exposure were significantly associated with ORS and zinc coutilisation (p≤0.05). The prevalence of ORS and zinc coutilisation was 1.14 (APR 1.14, 95% CI 1.07 o1.21), 1.13 (APR 1.13, 95% CI 1.06, 1.21) and 1.09 (APR 1.09, 95% CI 1.02 to 1.16) times higher in children whose mothers were aged 20–24, 25–29 and 30–34, respectively, in comparison to mother aged 15–19. Being the child of a woman with a primary, secondary or higher level of education raises the prevalence of coutilisation by 1.15, 1.08 and 1.19, respectively. Compared with children from poor household wealth index, those children from rich wealth index had a 4% higher prevalence of ORS and zinc coutilisation. The prevalence of ORS and zinc coutilisation among children whose mothers had media exposure was 1.04 times

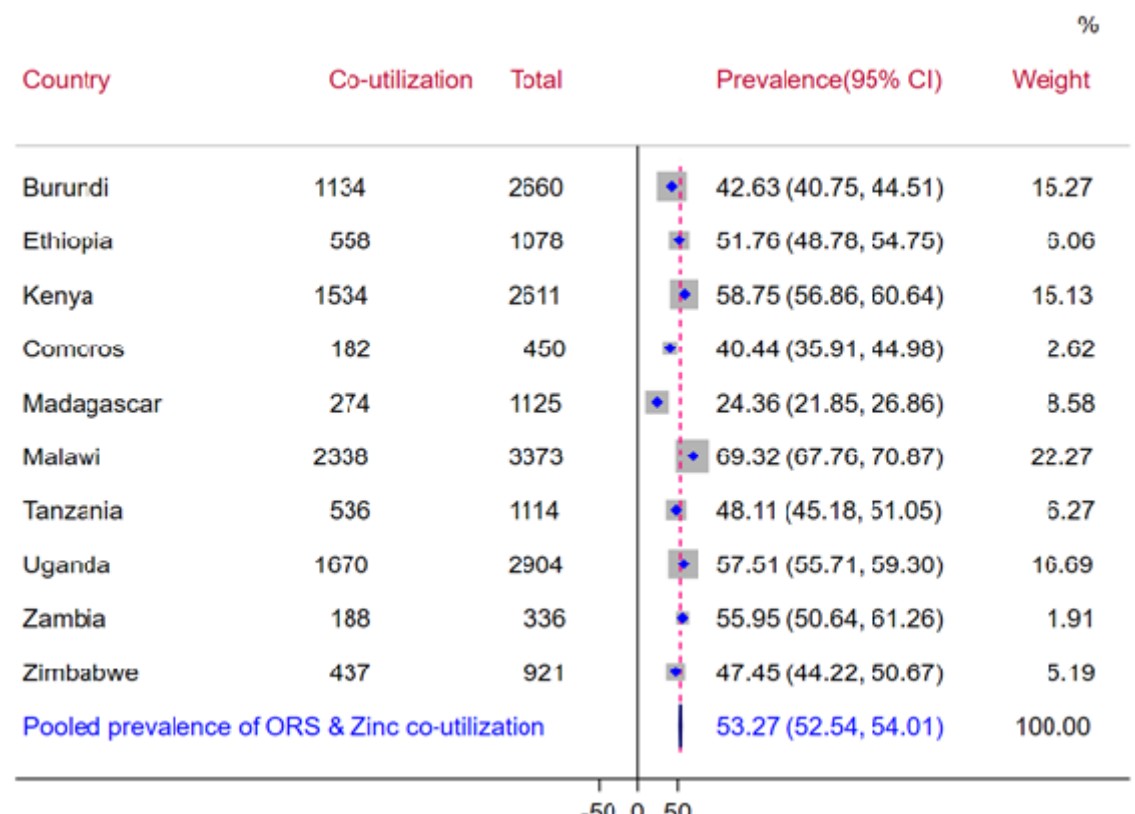

**Figure 1** Forest plot of pooled prevalence of coutilisation of ORS and zinc for management of diarrhoea among under-5 children in East Africa from 2011 to 2022. ORS, oral rehydration solution.

higher as compared with their counterparts (APR 1.04, 95% CI 1.01 to 1.08) (table 3).

## DISCUSSION

This study aimed to assess the factors associated with the co-use of ORS and zinc for the treatment of diarrhoea in under-5 children across East African countries. In our study, the co-use of ORS and zinc was associated with the mother's age, her educational level, family's financial status and media exposure.

This study mentioned maternal age above 20 is associated with coutilisation of ORS and zinc and this finding is supported by different study conducted in Nigeria.[26] The possible reason for this finding is older mothers were more likely than younger mothers to be knowledgeable about paediatric diarrhoea and how to treat it, younger mothers do not have as much exposure as older mothers about child diarrhoea and its management as maternal age can have impact on their knowledge about and how to treat it.[27] This study found that maternal educational status was associated with the usage of ORS with zinc to treat under-5 diarrhoeal disease and this finding is supported by studies conducted in Pakistan,[28] India,[29] Nigeria,[23] Cameron,[30] Kenya[31] and Ethiopia.[21 25 32–34] This may be because educated women are more likely to take sick children to the medical facility and a woman with education is more likely than a mother without education to take her ill kid for follow-up as well as to take the ordered medication as per recommended. In addition to the above reason, educated mothers have a better awareness regarding the health problems of their kids than a mother without education.[35 36]

**Table 2** Random effect and model comparison for factors associated with coutilisation of ORS and zinc for management of diarrhoea among under-5 children in East Africa from 2011 to 2022

|  | Null model | Model 1 | Model 2 | Model 3 |
|---|---|---|---|---|
| ICC | 14.56% | 15.31% | 15.01% | 15.01% |
| Log-likelihood | −14 402.21 | −14 369.22 | −14 398.24 | −14 367.59 |
| Deviance | 28 804.42 | 28 738.44 | 28 796.48 | 28 735.18 |

ICC, intraclass correlation coefficient; ORS, oral rehydration solution.

**Table 3** Factors associated with coutilisation of ORS and zinc for management of diarrhoea among under-5 children in East Africa from 2011 to 2022

| Variable | ORS and zinc coutilisation | | CPR (95% CI) | APR (95% CI) | P value |
|---|---|---|---|---|---|
| Individual-level variables | No (%) | Yes (%) | | | |
| Age of the mother | | | | | |
| 15–19 | 746 (52.03) | 687 (47.97) | 1 | 1 | <0.01 |
| 20–24 | 2106 (44.67) | 2609 (55.33) | 1.13 (1.06 to 1.2) | 1.14 (1.07 to 1.2)* | <0.01 |
| 25–29 | 2026 (44.57) | 2519 (55.43) | 1.12 (1.05 to 1.2) | 1.13 (1.06 to1.2)* | <0.01 |
| 30–34 | 1573 (48.73) | 1654 (51.27) | 1.06 (0.99 to 1.1) | 1.09 (1.02 to 1.2)* | 0.256 |
| 35–39 | 982 (49.99) | 982 (50.01) | 1.01 (0.94 to 1.1) | 1.04 (0.97 to 1.1) | 0.929 |
| 40–44 | 400 (52.90) | 357 (47.10) | 0.95 (0.87 to 1.0) | 1.00 (0.92 to 1.1) | 0.53 |
| 45–49 | 106 (51.32) | 100 (48.68) | 0.98 (0.85 to 1.1) | 1.05 (0.91 to 1.2) | |
| Maternal educational status | | | | | |
| No formal education | 2004 (54.67) | 1662 (45.33) | 1 | 1 | <0.01 |
| Primary | 3873 (45.33) | 4671 (54.67) | 1.17 (1.12 to 1.2) | 1.15 (1.09 to 1.2)* | <0.08 |
| Secondary | 1806 (45.81) | 2136 (54.19) | 1.14 (1.08 to 1.2) | 1.08 (1.02 to 1.1)* | <0.01 |
| Higher | 256 (36.75) | 440 (63.25) | 1.28 (1.19 to 1.4) | 1.19 (1.10 to 1.3)* | |
| Household wealth index | | | | | |
| Poor | 3819 (49.10) | 3958 (50.90) | 1 | 1 | 0.137 |
| Middle | 1511 (46.96) | 1707 (53.04) | 1.05 (1.01 to 1.1) | 1.03 (0.99 to 1.1) | 0.042 |
| Rich | 2609 (44.57) | 3245 (55.43) | 1.07 (1.03 to 1.1) | 1.04 (1.01 to 1.1)* | |
| Maternal employment status | | | | | |
| Not employed | 2769 (48.19) | 2977 (51.81) | 1 | 1 | 0.08 |
| Employed | 5170 (46.57) | 5933 (53.43) | 0.98 (0.95 to 1.1) | 0.97 (0.94 to 1.04) | |
| Media exposure | | | | | |
| No | 3022 (50.72) | 2936 (49.28) | 1 | 1 | 0.046 |
| Yes | 4917 (45.15) | 5974 (54.85) | 1.08 (1.04 to 1.1) | 1.04 (1.01 to 1.1)* | |
| Community illiteracy level | | | | | |
| Low | 1287 (42.96) | 1709 (57.04) | 1 | 1 | 0.065 |
| High | 6653 (48.02) | 7201 (51.98) | 0.92 (0.89 to 0.9) | 0.96 (0.92 to 1.03) | |
| Community poverty level | | | | | |
| Low | 3641 (47.32) | 4053 (52.68) | 1 | 1 | 0.095 |
| High | 4298 (46.95) | 4857 (53.05) | 0.99 (0.97 to 1.0) | 1.03 (0.99 to 1.1) | |

*p≤0.05.
APR, adjusted prevalence ratio; CPR, crude PR; ORS, oral rehydration solution.

Our study mentioned that media exposure was one of the factors associated with diarrhoeal treatment using ORS with zinc. This finding is in line with studies conducted in India,[29] Bangladesh,[37] Ghana[38] and Ethiopia.[25 34] This may be due to the media's important role in increasing the mother's or caregiver's understanding and awareness of the best ways to treat diarrhoea in children.[37]

As this study finds, a family's wealth status affects the usage of ORS with zinc. This finding is supported by different studies conducted in India[39] and Ethiopia.[25] This is due to the fact that low-income households in LMICs, such as those in Africa, particularly in East Africa, cannot afford the expenses of healthcare services, including ORS and zinc. Thus, when children from low-income

households get diarrhoea, they might not receive ORS with zinc.[40]

Our study was based on nationally representative large data and used appropriate statistical analysis techniques. Despite the aforementioned strengths, the measure of zinc utilisation practice was based on the mother's recall, which could lead to recall bias. Second, the DHS survey lacks information on zinc availability and the role of diet in zinc intake. As a result, we are unable to evaluate these factors. Additionally, some East African countries are not included in the DHS programme. Furthermore, due to the cross-sectional nature of the data, a clear temporality (cause-and-effect relationship) between the dependent and independent variables was not established.

## CONCLUSION

In this study, media exposure, rich household wealth status, maternal education and maternal age over 20 were associated factors for ORS and zinc co-use. To raise the prevalence of ORS and zinc coutilisation for the successful treatment of diarrhoea in children under the age of 5, it is advised to increase media exposure, maternal education and support the household wealth index. Using media as tool to advertise the importance of the coutilisation of ORS and zinc for all community will increase and boost the utilisation of the product. Empowering women through education also will raise the coutilisation of ORS and zinc. In addition, policies and intervention programmes that aim to stop unnecessary deaths of children under the age of 5 should be prioritised by preventing adolescent pregnancy.

**Author affiliations**
[1]Department of Pediatrics and Neonatal Nursing, Wollega University, Nekemte, Ethiopia
[2]Department of Pediatrics and Child Health Nursing, Ambo University College of Medicine and Public Health, Ambo, Ethiopia
[3]Department of Neonatal Health Nursing, University of Gondar College of Medicine and Health Sciences, Gondar, Ethiopia
[4]Department of Pediatrics and Child Health Nursing, Wolaita Sodo University, Sodo, Ethiopia
[5]Department of Public Health, College of Medicine and Health Sciences, Samara University, Samara, Ethiopia

**Acknowledgements** We express our gratitude to the MEASURE DHS for providing the study's dataset.

**Contributors** Conceptualisation: BTL and BLS. Data curation: BTL and BLS. Formal analysis: BLS and BTL. Investigation: BTL and BLS. Methodology: BLS and BTL. Software: BTL and BLS. Validation: BLS, BTL, WTW, GDG and YTW. Writing–original draft: BLS and BTL. Writing–review and editing: BLS, BTL, WTW, GDG and YTW. BTL is a guarantor of this study.

**Funding** The authors have not declared a specific grant for this research from any funding agency in the public, commercial or not-for-profit sectors.

**Competing interests** None declared.

**Patient and public involvement** Patients and/or the public were not involved in the design, or conduct, or reporting, or dissemination plans of this research.

**Patient consent for publication** Not applicable.

**Provenance and peer review** Not commissioned; externally peer reviewed.

**Data availability statement** Data are available on reasonable request. Through an online request to http://www.dhsprogram.com, permission to access the measure

demographic and health survey data used in this study was obtained. Data from the official program database are accessible to the public.

**ORCID iDs**
Bruck Tesfaye Legesse http://orcid.org/0000-0001-6639-3060
Wubet Tazeb Wondie http://orcid.org/0000-0003-1522-7905
Gezahagn Demsu Gedefaw http://orcid.org/0009-0003-3584-4545
Beminate Lemma Seifu http://orcid.org/0000-0002-9278-6826

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
