## [Reviewer comments · BMJ Open]

ARTICLE DETAILS

TITLE (PROVISIONAL)	Co-utilization of oral rehydration solution and zinc for treating diarrhea, and its associated factor among under-five children in East Africa: a multilevel robust Poisson regression
AUTHORS	Legesse, Bruck Tesfaye; Wondie, Wubet; Gedefaw, Gezahagn; Workineh, Yakob Tadese; Seifu, Beminate

VERSION 1 – REVIEW

REVIEWER	Gupta, Piyush University College of Medical Sciences
REVIEW RETURNED	17-Sep-2023

GENERAL COMMENTS	none
------

REVIEWER	Mohamed, Sagad University of Khartoum Faculty of Medicine
REVIEW RETURNED	05-Nov-2023

GENERAL COMMENTS	- Introduction, lines 50-60: giving information about the health situation in the context of East Africa would be appropriate and better than describing the situations in other countries like Bangladesh and Pakistan.- Methods, line 90: (based on the most recent Demographic and Health Survey (DHS) data): the authors should indicate the version dates clearly; i.e. whether they used the 2014 dataset, or the 2020 dataset...etc.- Methods, line 90: (data of 12 East African countries) These countries should be listed here.- Methods, line 101: (16,850 under-five children was considered for final analysis) it would be better to describe the exclusion process until they got this final number.- Methods, line 108: (independent variables): I guess that these variables were selected based on the availability of the data collected, as well as, the literature review. It would be better to mention that if my assumption was right.- Results, Table 3: Column (ORS and zinc co-utilization): the "yes" items are enough as the inverse ("no") can be easily deduced by the reader- Discussion: The authors should mention a little more discussion about how these independent variables contributed to the findings.- Limitations: Some east African countries are not included in the DHS program. This should be mentioned as a limitation to this study.- Conclusion: It would be better to emphasize the main reflection or implication of this study, rather than using sentences that focus on parts of the problem, such as ((to empower women through
---

	education, prevent teen pregnancy)) in the abstract and ((should be prioritised by preventing adolescent pregnancy)) in the conclusion section. The main implication of this study (to be stated) was presenting data and identifying factors that had significant impact on the ORS and zinc supplementation. Utilizing the identified factors could help target those vulnerable subgroups and improve accessibility and affordability of zinc and ORS. - References: there are some errors in the references. I noticed that all of them missed "journal name" and some of them contains errors in the authors' names (like ref 7, 11, 12, 13). They need to be revised and edited according to the journal instructions. - The manuscript although gets the meaning across, there are some writing issues. English language proofreading is needed. There are several sentences that need correction, editing, or rephrasing. For example: Line 40: (it is importance to...) - Line 82: (there is limited evidence no research) - Line 86: (is the first to its type). Also, The second paragraph in the discussion (starting from Line 203) needs editing and rewriting. the sentence (we are unable) in line 49 is better to be "we were unable" and (paediatric diarrhea) in line 206 is better to be "childhood diarrhea".
--	---

REVIEWER	Amil Dias, Jorge Hospital Lusíadas Porto, Paediatric Gastroenterology
REVIEW RETURNED	17-Nov-2023

GENERAL COMMENTS	This is an interesting study cover a large population, therefore even minor differences, as observed, may be regarded as robust. Results follow the expected direction, as education and wealth improve compliance with best practices in managing diarrhoea disease in children. I note that the final sentence in abstract ("... it is importance to empower women through education, prevent teen pregnancy.") mentions a topic (prevent teen pregnancy) that is not mentioned in the main manuscript. Perhaps the authors should consider correcting the discrepancy. As a minor note "the prevalence" in line 228 should be corrected.
---

REVIEWER	Rana, Juwel North South University, Department of Public Health
REVIEW RETURNED	24-Jan-2024

GENERAL COMMENTS	I would like to extend my gratitude for the opportunity to review this manuscript. Your work contributes to an important field of study, and I appreciate the effort that has gone into this research. However, I have a few observations and suggestions that might enhance the clarity and impact of your findings. Originality of Study: Upon reviewing the manuscript, I noticed similarities with previously published works. For instance, the study titled "Prevalence and predictors of oral rehydration therapy, zinc, and other treatments for diarrhoea among children under-five in sub-Saharan Africa" by Bright Opoku Ahinkorah et al., published on October 13, 2022 (https://doi.org/10.1371/journal.pone.0275495) Selam Fisiha Kassa 1, Tewodros Getaneh Alemu 1, Masresha Asmare Techane 1, Chalachew Adugna Wubneh 1, Nega Tezera Assimamaw 1, Getaneh Muluaem Belay 1, Tadesse Tarik Tamir 1, Addis Bilal Muhye 1, Destaye Guadie Kassie 1, Amare Wondim 1, Bewuketu Terefe 2, Bethelihem Tigabu Tarekegn 1, Mohammed
--

	Seid Ali 1, Beletech Fentie 1, Almaz Tefera Gonete 1, Berhan Tekeba 1, Bogale Kassahun Desta 1, Amare Demsie Ayele 1, Melkamu Tilahun Dessie 1, Kendalem Asmare Atalell 1 Interpretation of Study Variables: This uniform approach would aid in eliminating any potential confusion and maintain the clarity of your statistical analysis. Furthermore, the term 'community poverty status' seems to be equated with the household wealth index for each child. If this is intended to be a reflection of community-level factors, it may be helpful to clarify this in the text to avoid ambiguity. Model Construction: I would like to discuss the construction of the four models as mentioned in the manuscript. It is currently unclear whether a separate model was developed for each country or if all countries were included within each model using a unified modeling strategy. For enhanced clarity and robustness of the findings, it would be beneficial if the models incorporated all countries, taking into consideration the potential for country-level clustering. This approach is crucial for accurately capturing the nuances of the data across different national contexts. Additionally, authors did mention how they pooled the PRs, and meta-analysis was appropriate and necessary. While the manuscript references a forest plot for country-level estimates, I assumed that they meta-analyzed country-level estimates. This provides partial insight into country-level variation; a more detailed explanation of this aspect would greatly enhance the reader's understanding. Furthermore, the use of a multilevel model to estimate individual-level prevalence or prevalence ratio (PR) is intriguing. I would recommend a more comprehensive explanation of why this approach is particularly relevant and important in the context of your study. Such clarity would not only solidify the methodological foundation of the research but also potentially enhance its impact and applicability. Prevalence Ratio Explanation: The rationale behind using PR, given the use of prevalence or surveillance data from cross-sectional surveys, is partially explained. While the Odds Ratio (OR) is a commonly employed method in statistical analysis, it is particularly noteworthy for its application in case-control studies. It would be helpful to provide a more comprehensive explanation of what the PR indicates, whether it's a ratio or an incidence, to enhance the reader's understanding.
--	---

VERSION 1 – AUTHOR RESPONSE

Reviewer: #2(Dr. Sagad Mohamed, University of Khartoum Faculty of Medicine)

Concern 1: - Introduction, lines 50-60: giving information about the health situation in the context of East Africa would be appropriate and better than describing the situations in other countries like Bangladesh and Pakistan.

Author's response: Thank you for your concern. We used those countries' information to indicate the magnitude of the problem as they have relatively relateable diarrheal and diarrheal disease treatment gaps with East Africa. If it is inappropriate at all, we can amend/remove it. Thank you.

Concern 2: - Methods, line 90: (based on the most recent Demographic and Health Survey (DHS) data): the authors should indicate the version dates clearly; i.e. whether they used the 2014 dataset, or the 2020 dataset...etc.

Author's response: Thank you for your concern, we corrected it.

Concern 4: - Methods, line 90: (data of 12 East African countries) These countries should be listed here.

Author's response: we did it, thank you.

Concern 5: - Methods, line 101: (16,850 under-five children were considered for final analysis) It would be better to describe the exclusion process until they got this final number.

Author's response: Thank you, we did it. The study population was all under-five children with diarrhea within the last two weeks preceding the survey. So, those who didn't fulfill these criteria were excluded from the study.

Concern 6: - Methods, line 108: (independent variables): I guess that these variables were selected based on the availability of the data collected, as well as, the literature review. It would be better to mention that if my assumption was right.

Author's Response: Yeah, you are right that it is based on the availability of complete variables within all included East African DHS data.

Concern 7: - Results, Table 3: Column (ORS and zinc co-utilization): the "yes" items are enough as the inverse ("no") can be easily deduced by the reader

Author's Response: Thank you for your suggestion but we left it as it is to ease the confusion that may happen from the readers. But we can remove it if you believe that must be removed.

Concern 8: - Discussion: The authors should mention a little more discussion about how these independent variables contributed to the findings.

Author's response: We tried to add some points to the discussion points but we forgot to activate the track change option to show you. Thank you for your concern.

Concern 9: - Limitations: Some East African countries are not included in the DHS program. This should be mentioned as a limitation of this study.

Author's response: sure, thank you for your insightful comment and we added it to the limitation of the study.

Concern 10: - Conclusion: It would be better to emphasize the main reflection or implication of this study, rather than using sentences that focus on parts of the problem, such as ((to empower women through education, prevent teen pregnancy)) in the abstract and ((should be prioritized by preventing adolescent pregnancy)) in the conclusion section. The main implication of this study (to be stated) was presenting data and identifying factors that had a significant impact on the ORS and zinc supplementation. Utilizing the identified factors could help target those vulnerable subgroups and improve the accessibility and affordability of zinc and ORS.

Author's response: Thank you, we tried to add some more points as per your recommendation.

Concern 11: - References: there are some errors in the references. I noticed that all of them missed "journal name" and some of them contained errors in the authors' names (like ref 7, 11, 12, 13). They need to be revised and edited according to the journal instructions.

Author's response: We tried to correct it on the revised version of the manuscript using the EndNote 9 version, thank you.

Concern 12: - The manuscript although gets the meaning across, there are some writing issues. English language proofreading is needed. There are several sentences that need correction, editing, or rephrasing. For example: Line 40: (it is important to...) - Line 82: (there is limited evidence no research) - Line 86: (is the first to its type). Also, The second paragraph in the discussion (starting from Line 203) needs editing and rewriting. the sentence (we are unable) in line 49 is better to be "we were unable" and (pediatric diarrhea) in line 206 is better to be "childhood diarrhea".

Author's response: Thank you for your insightful comments, we improved on the revised version of the manuscript. We forgot to activate the track change on the word to show you how many errors we corrected but we tried to correct all English grammar and typing errors in the revised manuscript.

Reviewer: #3 (Dr. Jorge Amil Dias, Hospital Lusiadas Porto)

Concern 1: This is an interesting study cover a large population, therefore even minor differences, as observed, may be regarded as robust. Results follow the expected direction, as education and wealth improve compliance with best practices in managing diarrhoea disease in children. I note that the final sentence in abstract ("... it is importance to empower women through education, prevent teen pregnancy.") mentions a topic (prevent teen pregnancy) that is not mentioned in the main manuscript. Perhaps the authors should consider correcting the discrepancy.

Author's response: sure, thank you very much for your comments, and suggestions and we corrected it as per your recommendations.

Concern 2: As a minor note "the prevalence" in line 228 should be corrected.

Author's response: we corrected it, thank you.

Reviewer: #4 (Mr. Juwel Rana, North South University, University of Massachusetts Amherst)

Concern 1: I would like to extend my gratitude for the opportunity to review this manuscript. Your work contributes to an important field of study, and I appreciate the effort that has gone into this research. However, I have a few observations and suggestions that might enhance the clarity and impact of your findings.

Author's response: we are thankful for your kind words, and we have corrected points as per your suggestions and comments.

Concern 2: Originality of Study: Upon reviewing the manuscript, I noticed similarities with previously published works. For instance, the study titled "Prevalence and predictors of oral rehydration therapy, zinc, and other treatments for diarrhoea among children under-five in sub-Saharan Africa" by Bright Opoku Ahinkorah et al., published on October 13, 2022 (<https://doi.org/10.1371/journal.pone.0275495>)

Selam Fisiha Kassa 1, Tewodros Getaneh Alemu 1, Masresha Asmare Techane 1, Chalachew Adugna Wubneh 1, Nega Tezera Assimamaw 1, Getaneh Mulualem Belay 1, Tadesse Tarik Tamir 1, Addis Bilal Muhye 1, Destaye Guadie Kassie 1, Amare Wondim 1, Bewuketu Terefe 2, Bethelihem Tigabu Tarekegn 1, Mohammed Seid Ali 1, Beletech Fentie 1, Almaz Tefera Gonete 1, Berhan Tekeba 1, Bogale Kassahun Desta 1, Amare Demsie Ayele 1, Melkamu Tilahun Dessie 1, Kendalem Asmare Atalell 1

Author's Response: Thank you for your concern, the above studies are similar to our study only by the title. Regarding the first study you mentioned, it was conducted among Su-Saharan countries, and at the same time it is not about co-utilization rather it mentioned the treatment options and the prevalence of those treatment options utilization while our study was conducted among East African countries and solely on co-utilization of ORS and Zinc simultaneously for each under five child suffered diarrhea. The second study was conducted only on Ethiopian DHS data of a single country while our study is conducted among 12 East African countries.

Concern 3: Interpretation of Study Variables: This uniform approach would aid in eliminating any potential confusion and maintain the clarity of your statistical analysis. Furthermore, the term 'community poverty status' seems to be equated with the household wealth index for each child. If this is intended to be a reflection of community-level factors, it may be helpful to clarify this in the text to avoid ambiguity.

Author's response: Thank you for your comment, dear reviewer. We have aggregated wealth status at the EAs level to consider the neighborhood effect in the model because not only do individual's

characteristics affect their co-utilization but also the characteristics of individuals around them could influence it. We have incorporated your suggestion and added an operational definition for "community poverty level".

Community poverty level: The percentage of poor and poorest mothers in the cluster is specified. The proportion of poor and poorest women in each cluster was aggregated to reflect the overall poverty status within the cluster. Mothers were categorized based on their poverty level compared to the national median value.

Concern 4: Model Construction: I would like to discuss the construction of the four models as mentioned in the manuscript. It is currently unclear whether a separate model was developed for each country or if all countries were included within each model using a unified modeling strategy. For enhanced clarity and robustness of the findings, it would be beneficial if the models incorporated all countries, taking into consideration the potential for country-level clustering. This approach is crucial for accurately capturing the nuances of the data across different national contexts.

Author's response: Thank you for your comment. We have included all countries in the model and accounted for heterogeneity between them by using country-level clustering.

Concern 5: Additionally, the authors did mention how they pooled the PRs, and meta-analysis was appropriate and necessary. While the manuscript references a forest plot for country-level estimates, I assumed that they meta-analyzed country-level estimates. This provides partial insight into country-level variation; a more detailed explanation of this aspect would greatly enhance the reader's understanding.

Author's response:

Concern 6: Furthermore, the use of a multilevel model to estimate individual-level prevalence or prevalence ratio (PR) is intriguing. I would recommend a more comprehensive explanation of why this approach is particularly relevant and important in the context of your study. Such clarity would not only solidify the methodological foundation of the research but also potentially enhance its impact and applicability.

Author's response: Thank you for your comment, dear reviewer. This study was a cross-sectional study and the prevalence of ORS and Zinc co-utilization was greater than 10%, and if we report the odds ratio, it could overestimate the association between the dependent and the independent variables. In such cases, the prevalence ratio is the best measure of association, and therefore, multilevel Poisson regression analysis with robust variance was fitted to identify predictors of co-utilization.

Concern 7: Prevalence Ratio Explanation: The rationale behind using PR, given the use of prevalence or surveillance data from cross-sectional surveys, is partially explained. While the Odds Ratio (OR) is a commonly employed method in statistical analysis, it is particularly noteworthy for its application in

case-control studies. It would be helpful to provide a more comprehensive explanation of what the PR indicates, whether it's a ratio or an incidence, to enhance the reader's understanding.

Author's response: Thank you for your comment, dear reviewer. PR indicates the prevalence ratio.